# Risk of Hospital Readmission among Older Patients Discharged from the Rehabilitation Unit in a Rural Community Hospital: A Retrospective Cohort Study

**DOI:** 10.3390/jcm10040659

**Published:** 2021-02-09

**Authors:** Ryuichi Ohta, Chiaki Sano

**Affiliations:** 1Community Care, Unnan City Hospital, Iida, Daito-cho, Unnan, Shimane 699-1221, Japan; 2Department of Community Medicine Management, Faculty of Medicine, Shimane University, Izumo, Shimane 693-8501, Japan; sanochi@med.shimane-u.ac.jp

**Keywords:** rehabilitation, readmission, older people, geriatrics, body mass index, functional independence measure, activities of daily living, rural hospital

## Abstract

Rehabilitation for hospitalized older people can improve their independence for performing activities of daily living (ADL), but determining its appropriateness can be challenging because of inherent limitations in their ADL and short life expectancy. Thus, we aimed to clarify the benefit of rehabilitation among older Japanese patients. We retrospectively evaluated consecutive older patients (age > 65 years) admitted to the rehabilitation unit of a rural community hospital between 1 April 2016 and 31 March 2020. The primary outcome measure was readmission for acute conditions. Of the 732 patients evaluated, 311 patients (42.5%) were readmitted. Readmission was significantly associated with body mass index (BMI) (*p* < 0.001), dependent condition (*p* < 0.001), higher cognitive domain scores in the functional independence measure (FIM) (*p* = 0.019), and polypharmacy (*p* = 0.026). The most frequent cause of readmission was pyelonephritis (11.9%), followed by pneumonia (10.9%), compression fracture (10.6%), heat stroke (8.4%), and cerebral stroke (8.0%). In conclusion, older Japanese patients discharged from rehabilitation units have lower readmission rates than those previously reported. Thus, better nutritional control, a multidisciplinary approach to the management of cognitive dysfunction, and a decrease in polypharmacy could be associated with improved outcomes among discharged older patients.

## 1. Introduction

Rehabilitation is an important aspect of the continuity of home care among admitted older people as it can improve independence in their performance of activities of daily living (ADL) in their homes [1]. Rehabilitation increases the probability of home care and reduces long-term hospitalization [2,3,4]. The importance of home care in family medicine has increased with the rapid aging of the population worldwide [5,6]. However, home care can be challenging for older people in rural areas because of several barriers to the access to primary care [7,8]. Older people generally have several comorbidities; this places them at higher risk of hospital admission due to acute exacerbation of these conditions [9]. Successful discharge to home requires effective hospital rehabilitation and an interprofessional approach to rural home care [10,11]. As such, the outcomes of rehabilitation in rural community hospitals should be clarified.

A challenge in the rehabilitation of older patients is determining the appropriateness of continuing rehabilitation because of the possibility of a limited benefit in improving ADL and the short life expectancy in this population. Readmission to acute care hospitals is an established outcome measure of rehabilitation [12,13]. High functional status at discharge from rehabilitation centers helps lower the rate of readmission to acute care hospitals within 30 to 90 days [12,14,15]. Further, a higher ADL capability at discharge is a predictor of the success of home care. However, the effect of rehabilitation on ADL is lower in older people than that in younger people. Moreover, it is difficult to predict the benefits of rehabilitation in these individuals because of their frailty [16].

A longer duration of rehabilitation is related to a higher readmission rate [17,18]. Accordingly, the duration of rehabilitation should be decided through constant assessments of ADL improvements. In addition, older patients have a short life expectancy, and thus rehabilitation should involve the patient, the patient’s family, and a multidisciplinary team [18,19]. Older people with multimorbidity can still be readmitted with acute symptoms in the short term, despite undergoing rehabilitation [20,21]. Thus, the patient should be fully informed of the effect of rehabilitation and the planned home care duration.

The outcome of rehabilitation after discharge for older people is critical for the decision to continue rehabilitation. Most patients undergoing rehabilitation in rural community hospitals are aged over 80 years and are frail with multiple morbidities [10,22,23]. Rehabilitation should be personalized according to their home care goals and evidence of rural rehabilitation [24]. A previous study showed that improvement of the motor component of the functional independence measure (FIM) could contribute to home discharge in rural settings [10]. However, to our knowledge, no study has investigated the long-term effects of FIM and other patient demographics on the duration of home care in the rural setting. In addition, although home care duration can be related to health insurance policies and culture, data regarding these aspects are scarce in rural Japan. Furthermore, data regarding the outcomes of older people discharged after rehabilitation are limited. Therefore, this study aimed to clarify the duration of home care after discharge from a rehabilitation unit in a rural hospital. Specifically, we investigated the risk factors affecting the duration of home care and readmission among older Japanese patients discharged from a rehabilitation unit in a rural hospital.

## 2. Materials and Methods

### 2.1. Study Design and Patients

This was a retrospective cohort study of consecutive patients aged over 65 years who were admitted between 1 April 2016 and 31 March 2020 to Unnan City Hospital for acute diseases and transferred to the rehabilitation unit. Unnan City is a rural city located southeast of the Shimane Prefecture in Japan. In 2020, the total population was 37,638 (18,145 males and 19,492 females). Older individuals (age > 65 years) account for 39% of the population, and this is projected to increase to 50% by 2025 [25]. Unnan City had 16 clinics, 12 home care stations, 3 visiting nurse stations, and 1 public hospital (Unnan City Hospital) at the time of the study. The hospital staff involves 27 physicians, 197 nurses, 7 pharmacists, 15 clinical technicians, 37 therapists (22 physical therapists, 12 occupational therapists, and 3 speech therapists), 4 nutritionists, and 34 clerks. Unnan City has only one recovery rehabilitation unit. [10].

The patients were regularly followed up at Unnan City Hospital or other medical institutions in Unnan City from April 2016 to 30 October 2020. All readmissions were at Unnan City Hospital. All participants were followed until readmission or death (range, 180–1640 days).

### 2.2. Recovery Rehabilitation Unit

The recovery rehabilitation unit of Unnan City Hospital had 30 rehabilitation beds during the study period. The unit accommodated patients motivated to return home after rehabilitation. Most of the patients had underlying internal medicine or orthopedic conditions. The rehabilitation plan was discussed with the patients and their family by a physician and the chief nurse in charge of the recovery rehabilitation unit. The decision to move from acute care to the recovery rehabilitation unit was undertaken collaboratively. Rehabilitation was performed at an average of twice per day (60–90 min per session) by physical and occupational therapists. If the patient had swallowing and speaking problems, speech therapists were involved. The discharge timing and location were decided based on discussions among patients; their families; and a team consisting of the attending physician, nurse, and social workers. The team is specialized in providing support and decision-making for patient discharge [10].

### 2.3. Measurements

Patient information was extracted from the electronic medical records of Unnan City Hospital throughout the research period. The main outcome measure was readmission to the hospital after discharge from the rehabilitation unit. Planned readmissons for chemotherapy and surgeries were excluded. The following data were collected: age; sex; body mass index (BMI); serum albumin (g/dL) as an indicator of nutritional status; reasons for readmission; the number of medications used (to assess polypharmacy) [26]; the Charlson Comorbidity Index (CCI), which indicates the severity of the patient’s medical conditions [27]; the duration of home care after discharge; care level based on the Japanese long-term insurance system (rated from 1 to 5, with 1 indicating least dependence and 5, severe dependence); the cognitive and motor components of the FIM at discharge, which were measured by therapists as an indicator of patients’ ADL; and the places where the patients were discharged to (home or facility). The patients were divided into two groups: a readmission and a no readmission group. The reasons for admission were categorized as orthopedic and medicine-related conditions.

### 2.4. Statistical Analysis

Parametric data were compared using the Student’s t-test, while nonparametric data were analyzed using the Mann–Whitney U test. Based on previous studies and the average of variables, numerical variables were dichotomized as follows: CCI, ≥5 and <5 [27]; care level, ≥1 and <1, based on the burden on caregivers and families [28]; and number of medicines, ≥5 and <5. Polypharmacy was defined as taking >5 medicines [29]. The cognitive and motor component scores and the total score of the FIM at discharge (high and low) were dichotomized using the median of each variable because they were nonparametric data (31, 78, and 109, respectively). Variables reported to be significantly associated with discharge to home in previous studies were selected and analyzed [30,31,32]. Statistically significant factors in the univariate analysis were also entered into multivariate analysis with the Cox proportional hazard regression model to determine independent predictors of readmission after discharge. Cumulative event-free survival rates were calculated using the Kaplan–Meier method and analyzed using the log-rank test. Cases with missing data were excluded from the analysis. All statistical analyses were performed using EZR version 1.51(Saitama Medical Center, Jichi Medical University, Saitama, Japan) version 1.51, which is a graphical user interface for R (The R Foundation, Vienna, Austria) [33]. A *p*-value of <0.05 was considered statistically significant.

### 2.5. Ethical Considerations

The hospital was assured of no loss of anonymity and confidentiality regarding the patients′ information. The information related to this study was posted on the hospital website without the disclosure of any details concerning the patients. To address any questions regarding this study, contact information of the hospital representative was also listed on the website. The Unnan City Hospital Clinical Ethics Committee approved this study (protocol code 20200023; date of approval: August 2020).

## 3. Results

### 3.1. Patient Characteristics

Of the 951 patients admitted to the rehabilitation unit, 845 patients were aged >65 years. After excluding 113 patients with missing data, 732 participants were finally evaluated. The patient inclusion flowchart is shown in Figure 1.

The average patient age was 84 years (standard deviation = 8.06), and 32.7% of the participants were male. A total of 311 participants (42.5%) were readmitted to the hospital. BMI (*p* < 0.001), number of medicines (*p* = 0.014), CCI (*p* = 0.004), and dependent condition (*p* < 0.001) were higher in the readmission group.

Meanwhile, albumin concentration was higher in the non-readmission group (*p* = 0.003). Regarding FIM, all scores were statistically higher in the non-readmission group. The length of rehabilitation was longer in the readmission group (Table 1).

With respect to the timing of readmission, 25.54% of the participants were admitted to the hospital within 365 days (Table 2). The interval between discharge and readmission ranged from 181 days to 730 days (Figure 2).

### 3.2. Regression Model Results

Kaplan–Meier curves show the estimated probability of non-readmission as a function of total FIM and of the cognitive and motor components of FIM (Figure 2). A Cox regression analysis was performed with age, albumin, BMI, CCI ≥ 5, dependent condition, discharge to home, higher cognitive and motor domain scores, polypharmacy, and length of rehabilitation. The results are presented in Table 3. BMI (*p* < 0.001), dependent condition (*p* < 0.001), higher cognitive domain scores of FIM (*p* = 0.019), and polypharmacy (*p* = 0.026) were significantly associated with readmission.

### 3.3. Reasons for Readmission to the Hospital

Table 4 lists the diagnoses and the frequencies for hospital readmission. The most frequent cause of readmission was pyelonephritis (11.9%) followed by pneumonia (10.9%), compression fracture (10.6%), heat stroke (8.4%), and cerebral stroke (8.0%).

## 4. Discussion

Determination of the benefits of rehabilitation in older patients can be challenging because of the inherent limitations in their ADL and their short life expectancy. In this study, less than 50% of the patients who underwent rehabilitation were readmitted to the hospital within 4 years. Lower BMI, dependent conditions, lower cognitive domain scores in FIM, and polypharmacy were related to readmission after discharge from the rehabilitation unit. The most frequent causes of readmission were infections, including pyelonephritis and pneumonia.

The lower readmission rate for patients admitted to rural hospitals can be partly attributed to the Japanese rehabilitation system and health insurance. The 30- and 90-day readmission rates in this study were much lower than those in previous studies [34,35,36,37], and this may be related to the duration of rehabilitation. Previous studies, which were mostly performed in the United States, reported a rehabilitation period of 20 days [34,35,36,37], which is lower than the 52 days in this study. Furthermore, there are differences in healthcare systems between the United States and Japan [38]. The quality of hospitals in the United States is assessed by the duration of hospital stay; thus, older patients have to be discharged from the hospital early [39]. In contrast, the rehabilitation of older patients is supported by social insurance in Japan; thus, the cost is mostly compensated by the government [38]. Older Japanese people can undergo adequate rehabilitation to achieve a better FIM at discharge. In turn, this can prevent early readmission.

We also found that a higher BMI was associated with a lower risk of readmission. In older patients, a higher BMI can indicate better nutrition and muscle conditions. The relationship between BMI and mortality indicates that a higher BMI can result in better longevity [40]. Previous studies reported sarcopenia among obese patients, which indicates that a high BMI may not immediately indicate more muscle [41,42,43]. Our results showed that older rural people with a higher BMI, based on an average value of 22 kg/m^2^, had lower readmission rates. Accordingly, this population should aim to have a moderated BMI that is ≥22 kg/m^2^. To achieve better outcomes in rehabilitation units, the healthcare staff should also include the patient’s BMI in the rehabilitation plan, ensuring that patients with BMIs lower than 22 kg/m^2^ are provided interventions to increase their BMI through training and better nutrition.

Cognitive function at discharge and dependent condition were also identified as influencing factors for readmission. Low cognitive function can mask critical symptoms, leading to disease exacerbation and admission. Good cognitive function is important in older patients because it will enable them to effectively convey their symptoms. Some dependent older people with low cognitive function are isolated in rural areas; thus, home care professionals struggle to manage older people who cannot accurately report their symptoms [44,45,46]. Furthermore, rural areas lack healthcare resources; as such, a multidisciplinary education program is important for home care patients [47]. However, miscommunication regarding symptoms can limit collaboration among home care professionals [48]. The patient’s cognitive function and risk of readmissions, as well as improvements in cognitive FIM, need to be assessed before discharge from the rehabilitation unit [49]. As shown in this study, 13.9% of the readmitted patients had dehydration or a heat stroke, which can lead to cognitive dysfunction in older individuals [50]. Therefore, the water intake of older patients should be carefully monitored so that dehydration is detected in the early stage and the risk of readmission is reduced.

Polypharmacy is implemented before discharge in order to reduce the risk of readmission. Polypharmacy, which is common among older patients, can affect the physiology of older patients [51] and cause femoral and compression fractures as well as cognitive dysfunction [26,29,52]. In this study, one of the frequent causes of readmission was fracture, which can be caused by the effects of polypharmacy. Reducing the number of medications during rehabilitation can lower the risk of readmission among older patients. A previous study showed that discontinuing sleep and depressive drugs can improve cognitive function in this population [53]. As polypharmacy can be easily missed during acute treatment, healthcare professionals should carefully assess the number of medications taken by the patient and implement active measures to lower polypharmacy after shared decision-making with the patients and their families [54,55,56].

This study has some limitations. First, it was performed in a single rehabilitation center in a rural community hospital, and this may have affected the external validity. Future studies should investigate the outcomes of older patients in different types of hospitals in other countries. The second limitation was the low follow-up rate. Some patients were discharged to other cities and could not be followed-up; this affected the reliability of our findings. Despite these limitations, our findings provide data that may also be reflective of other settings in Japan. The data may be used as a base for developing relevant guidelines on the rehabilitation of older patients, including those in other countries, and prevent readmission.

## 5. Conclusions

A higher BMI, no dependent conditions, a higher cognitive domain score in FIM, and no polypharmacy could be associated with the risk of readmission among older patients admitted to rehabilitation units. Thus, better nutritional control, collaboration among healthcare professionals for the management of cognitive dysfunction, and efforts to reduce polypharmacy are important for improving the outcomes of these patients.

## Figures and Tables

**Figure 1 jcm-10-00659-f001:**
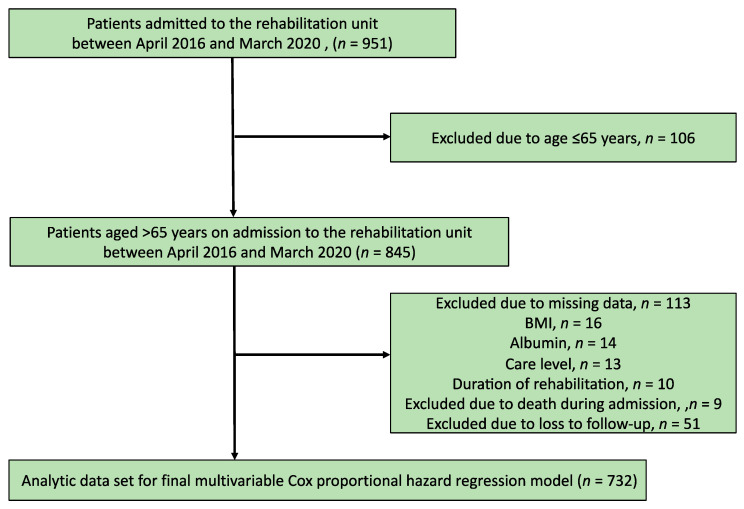
Patient inclusion flowchart.

**Figure 2 jcm-10-00659-f002:**
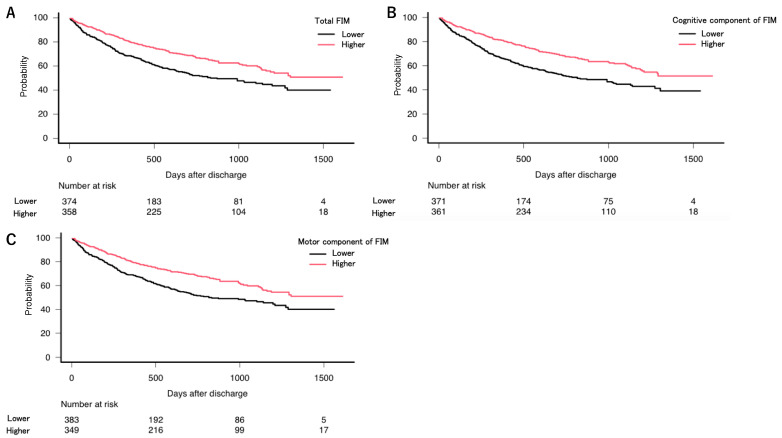
Kaplan–Meier curves showing the probability of non-readmission after discharge. Graphs are presented for total FIM (**A**) and cognitive (**B**) and motor (**C**) components of FIM.

**Table 1 jcm-10-00659-t001:** Patient characteristics.

		Readmission		
Factor		Yes	No	*p* Value
*n*	732	311	421	
Readmission (%)	311 (42.5)	311 (100.0)	0 (0.0)	<0.001
Days from discharge, mean (SD)	647.52 (438.84)	368.35 (320.62)	853.75 (398.71)	<0.001
Age, mean (SD)	84.00 (8.06)	85.58 (7.45)	82.83 (8.30)	<0.001
Male sex (%)	239 (32.7)	95 (30.5)	144 (34.2)	0.301
Albumin, mean (SD)	3.74 (0.56)	3.66 (0.59)	3.79 (0.53)	0.003
BMI, mean (SD)	21.40 (4.96)	20.65 (3.87)	21.99 (5.60)	<0.001
CCI ≥ 5 (%)	447 (61.1)	209 (67.2)	238 (6.5)	0.004
CCI (%)				
1	5 (0.7)	3 (0.9)	2 (0.4)	
2	17 (2.3)	7 (2.3)	10 (2.4)	
3	57 (7.8)	16 (5.1)	41 (9.7)	
4	206 (28.1)	76 (24.4)	130 (30.9)	
5	154 (21.0)	63 (20.3)	91 (21.6)	
6	148 (20.2)	69 (22.2)	79 (18.8)	
7	78 (10.7)	41 (13.2)	37 (8.8)	
8	37 (5.1)	21 (6.8)	16 (3.8)	
9	20 (2.7)	12 (3.9)	8 (1.9)	
10	9 (1.2)	3 (1.0)	6 (1.4)	
12	1 (0.1)	0 (0.0)	1 (0.2)	
Heart failure (%)	97 (13.3)	46 (14.8)	51 (12.1)	0.321
Asthma (%)	34 (4.6)	16 (5.1)	18 (4.3)	0.598
Kidney diseases (%)	143 (19.5)	62 (19.9)	81 (19.2)	0.851
Liver diseases (%)	24 (3.2)	13 (4.2)	11 (2.6)	0.282
COPD (%)	24 (3.3)	9 (2.9)	15 (3.6)	0.679
DM (%)	133 (18.2)	54 (17.4)	79 (18.8)	0.698
Brain hemorrhage (%)	84 (11.5)	34 (10.9)	50 (11.9)	0.726
Brain infarction (%)	163 (22.3)	72 (23.2)	91 (21.6)	0.654
Hemiplegia (%)	26 (3.6)	10 (3.2)	16 (3.8)	0.84
Dementia (%)	65 (8.9)	29 (9.3)	36 (8.6)	0.793
Connective diseases (%)	35 (4.8)	16 (5.1)	19 (4.5)	0.728
Cancer (%)	124 (16.9)	55 (17.7)	69 (16.4)	0.797
Dependent condition (%)	210 (28.7)	121 (38.9)	89 (21.1)	<0.001
Care level (%)				
0	522 (71.3)	190 (61.1)	332 (78.9)	
1	39 (5.3)	17 (5.5)	22 (5.2)	
2	71 (9.7)	43 (13.8)	28 (6.7)	
3	47 (6.4)	25 (8.0)	22 (5.2)	
4	28 (3.8)	17 (5.5)	11 (2.6)	
5	25 (3.4)	19 (6.1)	6 (1.4)	
Reason for admission				
Medicine-related	393 (53.7)	163 (52.4)	230 (54.6)	0.600
Orthopedic	339 (46.3)	148 (47.6)	191 (45.4)	
Locations of discharge (%)				
Nursing facility	129 (17.6)	57 (18.3)	72 (17.1)	0.695
Home	603 (82.4)	254 (81.7)	349 (82.9)	
Number of medicines taken, mean (SD)	5.89 (2.36)	6.14 (2.47)	5.70 (2.26)	0.014
Number of patients with polypharmacy, *n* (%)	538 (73.5)	240 (77.2)	298 (70.8)	0.062
FIM score at discharge				
Total FIM score (median)	109 (18, 126)	104(18, 126)	111 (18, 126)	0.005
Motor domain score (median)	31.00 (13, 35)	30.00 (13, 35)	32.00 (13, 35)	0.014
Cognitive domain score (median)	78 (5, 91)	74 (5, 91)	79 (13, 91)	0.005
Duration of rehabilitation (median)	52 (3, 228)	49 (5, 228)	57(3, 189)	0.015

CCI, Charlson Comorbidity Index CCI; SD, standard deviation; COPD, chronic obstructive pulmonary disease; DM, diabetes mellitus; FIM, functional independence measure.

**Table 2 jcm-10-00659-t002:** Interval between discharge and readmission (days).

Factor	*n* = 732	Percentage	Cumulated Percentage
Interval	311	42.49%	
<30 days	23	3.14%	3.14%
30 to 90 days	46	6.28%	9.42%
91 to 180 days	39	5.33%	14.75%
181 to 365 days (1 year)	79	10.79%	25.54%
366 to 730 days (2 years)	80	10.93%	36.47%
731 to 1095 days (3 years)	28	3.83%	40.30%
>1096 days	16	2.19%	42.49%

**Table 3 jcm-10-00659-t003:** Results of the Cox regression model for hospital readmission.

Factor	Hazard Ratio	95% CI	*p* Value
Age	1.01	0.99–1.02	0.49
Male sex	0.84	0.65–1.08	0.16
Albumin	0.89	0.72–1.10	0.28
BMI	0.95	0.92–0.98	<0.001
CCI ≥ 5	1.21	0.94–1.57	0.14
Dependent condition	2.01	1.56–2.59	<0.001
Discharge to home	1.08	0.78–1.49	0.64
FIM score			
Higher cognitive domain score	0.71	0.53–0.94	0.019
Higher motor domain score	0.93	0.68–1.27	0.66
Polypharmacy	1.36	1.04–1.79	0.026
Length of rehabilitation	1	0.99–1.00	0.077

**Table 4 jcm-10-00659-t004:** Frequency of readmission by cause.

Diagnosis	Number	Percentage	Diagnosis	Number	Percentage
Pyelonephritis	37	11.9%	Pseudogout	6	1.9%
Pneumonia *	34	10.9%	Ileus	6	1.9%
Compression fracture	33	10.6%	Epilepsy	5	1.6%
Heat stroke	26	8.4%	Liver failure	4	1.3%
Cerebral stroke *	25	8.0%	Renal failure	3	1.0%
Other infections *	24	7.7%	LSS	3	1.0%
Femoral fracture	22	7.1%	Peripheral vertigo	2	0.6%
Other fractures *	19	6.1%	Osteoarthritis	2	0.6%
Trauma	17	5.5%	Asthma	2	0.6%
Dehydration	16	5.1%	Peptic ulcer*	2	0.6%
Cancer *	11	3.5%	Ischemic colitis	1	0.3%
Autoimmune diseases *	10	3.2%	Angina	1	0.3%

* Pneumonia includes bacterial, viral, and aspiration pneumonia. Cerebral strokes included brain infarction, brain hemorrhage, and TIA (Transient ischemic attack). Other infections include cholecystitis, cholangitis, septic arthritis, cellulitis, diverticulitis, appendicitis, psoas abscess, tuberculosis, and hepatic abscess. Other fractures include fractures to the pelvis, foot, rib, arm, and clavicle. Trauma includes head, elbow, and knee injuries. Cancer includes colon, stomach, pancreas, bone, and brain cancers. Autoimmune diseases include rheumatoid arthritis, polymyalgia rheumatica, and temporal arthritis. Peptic ulcers include gastric and duodenal ulcers.

## Data Availability

The datasets used and/or analyzed during the current study are available from the corresponding author upon reasonable request.

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
