# Peer review of "Risk of Hospital Readmission among Older Patients Discharged from the Rehabilitation Unit in a Rural Community Hospital: A Retrospective Cohort Study"

_jcm, 2021, doi:10.3390/jcm10040659_

Round 1
Reviewer 1 Report
Abstract - a structured abstract format would improve clarity
Introduction
The introduction is sufficiently broad to support the research question and frames the problem well.
Methods
Methods are appropriate and conventional. No controversy with methods or analysis. Ethical oversight documented later in manuscript. Consider adding a section on ethical approval and oversight to the methods section.
Results
Tables have significant formatting issues that must be corrected - mixed text vertical and horizontal and overlapping text. Unsuitable in current form.
In the text, results are presented well.
Discussion
Conclusions are overstated. Example - Higher BMI is associated with a lower risk of readmission based on the data, the discussion states that a higher BMI is causal for a lower risk of readmission. The authors should use caution when evaluating the results and differential clearly between association and causation.
Conclusions
Overstated (see discussion above).
Author Response
Abstract - a structured abstract format would improve clarity.
Response:
Thank you for the input. However, we followed the journal guidelines, which state “The abstract should be a total of about 200 words maximum. The abstract should be a single paragraph and should follow the style of structured abstracts, but without headings”. If you still prefer that we structure the Abstract, do let us know, and we will be happy to add the subheadings Background, Methods, Results, and Conclusion.
Introduction
The introduction is sufficiently broad to support the research question and frames the problem well.
Response:
Thank you for the positive feedback.
Methods
Methods are appropriate and conventional. No controversy with methods or analysis. Ethical oversight documented later in manuscript. Consider adding a section on ethical approval and oversight to the methods section.
Response:
Thank you for the positive feedback. We have added a paragraph describing the ethical considerations in the Materials and Methods section.
Results
Tables have significant formatting issues that must be corrected - mixed text vertical and horizontal and overlapping text. Unsuitable in current form.
In the text, results are presented well.
Response:
We apologize for the poor table format. We have revised Tables 1 and 2 for better readability.
Discussion
Conclusions are overstated. Example - Higher BMI is associated with a lower risk of readmission based on the data, the discussion states that a higher BMI is causal for a lower risk of readmission. The authors should use caution when evaluating the results and differential clearly between association and causation.
Response:
Thank you for pointing this out. In accordance with your suggestions, we have toned down the conclusions in the revised manuscript.
Conclusions
Overstated (see discussion above).
Response:
Thank you for pointing this out. In accordance with your suggestions, we have toned down the conclusions in the revised manuscript.
Reviewer 2 Report
Thank you for the opportunity to participate in the manuscript review process for this article. I congratulate the authors on their hard work in this important area and look forward to its ultimate publication. The topic of readmissions reduction after acute rehabilitation is important and this article brings forward original knowledge to share. Below are my thoughts and suggested changes in the major manuscript review categories
Research question: Lines 69-72 - The research question as framed at the end of the introduction is “does functional Independence measure (FIM) affect the ability to be at home (e.g. reduce readmissions) for older adults?”. The research question is actually broader than this as described in the article. This article was not just about FIM. There was a univariate analysis followed by multivariate analysis looking at many variables associated with readmissions. I believe therefore the description of the research question needs to change to reflect this is not just about FIM.
Relevance: This article is relevant to the journal’s mission. The question of contributing factors to readmissions of patients undergoing acute rehabilitation care is important because of the aging population and need to appropriately allocate resources. The article does a very good job in summarizing recent research and knowledge. The main gap that is identified in the introduction is the difficulty in predicting who will benefit from rehabilitation given possible limited benefit because of short life expectancy of elderly patients. The article therefore tries to fill in the knowledge gaps as described and therefore there is originality to this article.
Research design (Materials & Methods): This is a retrospective cohort study of consecutively admitted patients over a 4-year period from 2016-2020 of patient's age 65 and older to an acute rehabilitation unit. The patients are followed up to 7 months beyond the study period. Data was collected from the electronic medical record. Data included demographic information, readmission information, medical and diagnostic information. The researchers then examined 2 groups of patients, those that were readmitted versus not readmitted during the study period. Out of 951 total admissions to the rehabilitation unit after excluding for age and missing data the study team was left with 732 patients of which 311 were readmitted. Main outcome was readmission to the hospital after discharge from the acute rehabilitation unit. The article gives enough detail that the study could be replicated by others. They did a univariant analysis looking at factors that are associated with readmissions and then a Cox regression multivariate analysis to determine hazard ratios for these variables.
Questions that need to be addressed:
- Confounding variable that needs to be addressed prior to publication. The patient group included a mix of medicine patients and orthopedic patients. Orthopedic patients could potentially have lower rates of readmissions and healthcare issues driving readmissions because their index hospitalization and rehabilitation stay was for defined limited orthopedic issue (e.g. elective hip/knee replacement). Also many orthopedic patients have been risk stratified prior to surgery to limit healthcare risk for issues such as readmissions. Patients admitted to acute rehabilitation because of medical issues may be more complicated. Therefore I suggest either analyzing orthopedic versus medicine patients and their readmission rates and impact to this study or at least acknowledging this potential confounding variable in the article.
- Line 107-117 Please clarify whether data that’s is collected from EMR is from the index hospitalization, rehabilitation stay, or throughout the four year study period
- Line 89-90: Please list criteria for defining readmissions. Were there any exclusions? For example sometimes psychiatric illness, oncology illnesses, planned readmissions for surgeries, etc. are excluded in the definition of readmission. Were these readmissions same site (Unnan hospital) only?
- Lines 307-309: Please clarify whether readmissions could have occurred to other hospitals and artifactually lowered the readmission rate. Article states there may have been a low follow up rate. How was this measured? What data supports this statement?
Results, Data analysis, and statistics: All variables were dichotomized and there was a univariate analysis followed by Cox regression multivariate analysis that showed the following variables were associated with readmission risk: BMI, dependent condition, cognitive score, polypharmacy. Additionally, the article describes the distribution of readmissions by disease causation. There were adequate number of patients to provide statistical significance in the variables they reviewed. The article reveals the statistical software package used for the analysis. If there are further questions regarding the statistical analysis used then this should be referred to a peer reviewer with advanced training in statistics. Please clarify or address the following:
- Table 1 is unreadable in the pdf version I have been sent
- Table 2 and Figure 2 state the same thing and showing two is superfluous. Therefore should choose one or the other for publication.
Discussion & Conclusion: The article’s discussion section describes the variables that they examined and found were associated with readmissions as well as the medical causes of those readmissions. The article then appropriately discusses why these variables may be associated with readmissions and references appropriately other literature that supports the hypothesis around these linkages. The following areas need to be addressed in the discussion section:
- Line 263-265: This is an inaccurate generalization. CMS provides much financial support for the cost of rehabilitation. I suspect there are major differences between Japan and United States in length of stay of both inpatient hospital stay and acute rehabilitation stay that are the differences that account for readmission rate differences.
- Discussion section would be easier to read and process if there were 4 separate paragraphs for each of the main variables you are discussing: BMI, Cognitive function, polypharmacy, & dependency
- Line 274: The article states "our result shows that rural older people with higher BMI have better nutritional conditions" is not correct. Albumin is a very poor predictor of malnutrition. This study was not designed to prove that hypothesis. Please rephrase this.
- Line 288: The article states "most patients who are readmitted had dehydration". I do not believe this is accurate as the table for only indicates 5.1% of all readmissions were from dehydration. Please consider rephrasing this.
Title, author list, abstract, ethics, references:
Title: Title of the article should be changed to reflect the singular hospital that this study involved. It says "risk of hospital readmission among older patients discharge from rehabilitation units in rural community hospitals". There was only one hospital involved in the study.
Abstract:
- Line 25-27: The article needs to be clear that this type of study cannot differentiate between causation and correlation. The abstract states that the variables that they discovered "helped lower the rate of readmission" and that addressing these variables "will help improve outcomes". This is similar in the conclusion line 315 in which the article says the 4 variables “lower the risk of readmission”. Although I share the optimism that addressing these variables is the right approach the language needs to be changed to reflect this is a correlation rather than a causation.
- Keywords: Consider adding more keywords to help with search engine optimization: geriatrics, BMI, FMI, ADL?
Ethics: Line 329-330 Informed consent statement states that informed consent was obtained from all subjects involved in the study. Is this a correct statement? That would mean there would have to be written informed consent from all 731 participants. Maybe that is part of routine admission to this hospital (obtaining a blanket informed consent statement for research) but usually one needs IC done separately from clinical hospital treatment release.
Author Response
Thank you for the opportunity to participate in the manuscript review process for this article. I congratulate the authors on their hard work in this important area and look forward to its ultimate publication. The topic of readmissions reduction after acute rehabilitation is important and this article brings forward original knowledge to share. Below are my thoughts and suggested changes in the major manuscript review categories
Research question: Lines 69-72 - The research question as framed at the end of the introduction is “does functional Independence measure (FIM) affect the ability to be at home (e.g. reduce readmissions) for older adults?”. The research question is actually broader than this as described in the article. This article was not just about FIM. There was a univariate analysis followed by multivariate analysis looking at many variables associated with readmissions. I believe therefore the description of the research question needs to change to reflect this is not just about FIM.
Response:
Thank you for the excellent insights. We have now restructured the description of the research question and study aims in the revised manuscript.
Relevance: This article is relevant to the journal’s mission. The question of contributing factors to readmissions of patients undergoing acute rehabilitation care is important because of the aging population and need to appropriately allocate resources. The article does a very good job in summarizing recent research and knowledge. The main gap that is identified in the introduction is the difficulty in predicting who will benefit from rehabilitation given possible limited benefit because of short life expectancy of elderly patients. The article therefore tries to fill in the knowledge gaps as described and therefore there is originality to this article.
Response:
Thank you very much for the positive feedback and appreciation.
Research design (Materials & Methods): This is a retrospective cohort study of consecutively admitted patients over a 4-year period from 2016-2020 of patient's age 65 and older to an acute rehabilitation unit. The patients are followed up to 7 months beyond the study period. Data was collected from the electronic medical record. Data included demographic information, readmission information, medical and diagnostic information. The researchers then examined 2 groups of patients, those that were readmitted versus not readmitted during the study period. Out of 951 total admissions to the rehabilitation unit after excluding for age and missing data the study team was left with 732 patients of which 311 were readmitted. Main outcome was readmission to the hospital after discharge from the acute rehabilitation unit. The article gives enough detail that the study could be replicated by others. They did a univariant analysis looking at factors that are associated with readmissions and then a Cox regression multivariate analysis to determine hazard ratios for these variables.
Response:
Thank you for the detailed review and positive feedback regarding our manuscript.
Questions that need to be addressed:
- Confounding variable that needs to be addressed prior to publication. The patient group included a mix of medicine patients and orthopedic patients. Orthopedic patients could potentially have lower rates of readmissions and healthcare issues driving readmissions because their index hospitalization and rehabilitation stay was for defined limited orthopedic issue (e.g. elective hip/knee replacement). Also many orthopedic patients have been risk stratified prior to surgery to limit healthcare risk for issues such as readmissions. Patients admitted to acute rehabilitation because of medical issues may be more complicated. Therefore I suggest either analyzing orthopedic versus medicine patients and their readmission rates and impact to this study or at least acknowledging this potential confounding variable in the article.
Response:
Thank you for the valuable recommendations. We categorized the reasons for readmission as orthopedic and medicine-related conditions and have shown their comparison in Table 1.
- Line 107-117 Please clarify whether data that’s is collected from EMR is from the index hospitalization, rehabilitation stay, or throughout the four year study period
Response:
We apologize for the lack of clarity regarding the data collection period. Patient information was extracted from the electronic medical records of Unnan City Hospital throughout the research period. We have specified this in the revised manuscript.
- Line 89-90: Please list criteria for defining readmissions. Were there any exclusions? For example sometimes psychiatric illness, oncology illnesses, planned readmissions for surgeries, etc. are excluded in the definition of readmission. Were these readmissions same site (Unnan hospital) only?
Response:
Thank you for the pertinent questions. The main outcome measure was readmission to the hospital after discharge from the rehabilitation unit. Planned readmissons for chemotherapy and surgeries were excluded. The patients were regularly followed up at Unnan City Hospital or other medical institutions in Unnan City from April 2016 to October 30, 2020. All readmissions were at Unnan City Hospital. We have specified these details in the revised manuscript.
- Lines 307-309: Please clarify whether readmissions could have occurred to other hospitals and artifactually lowered the readmission rate. Article states there may have been a low follow up rate. How was this measured? What data supports this statement?
Response:
Thank you for the pertinent questions. The patients were regularly followed up at Unnan City Hospital or other medical institutions in Unnan City from April 2016 to October 30, 2020. Patients were consulted at or transferred to Unnan City Hospital when they had acute symptoms. We have specified this in the revised manuscript and added the number of patients excluded due to loss of follow-up in Figure 1.
Results, Data analysis, and statistics: All variables were dichotomized and there was a univariate analysis followed by Cox regression multivariate analysis that showed the following variables were associated with readmission risk: BMI, dependent condition, cognitive score, polypharmacy. Additionally, the article describes the distribution of readmissions by disease causation. There were adequate number of patients to provide statistical significance in the variables they reviewed. The article reveals the statistical software package used for the analysis. If there are further questions regarding the statistical analysis used then this should be referred to a peer reviewer with advanced training in statistics. Please clarify or address the following:
- Table 1 is unreadable in the pdf version I have been sent
Response:
We apologize for the poor clarity of Table 1. We have revised the format of the table such that it is readable.
- Table 2 and Figure 2 state the same thing and showing two is superfluous. Therefore should choose one or the other for publication.
Response:
Thank you for the helpful suggestion. We have deleted Figure 2 and renumbered the figures.
Discussion & Conclusion: The article’s discussion section describes the variables that they examined and found were associated with readmissions as well as the medical causes of those readmissions. The article then appropriately discusses why these variables may be associated with readmissions and references appropriately other literature that supports the hypothesis around these linkages. The following areas need to be addressed in the discussion section:
- Line 263-265: This is an inaccurate generalization. CMS provides much financial support for the cost of rehabilitation. I suspect there are major differences between Japan and United States in length of stay of both inpatient hospital stay and acute rehabilitation stay that are the differences that account for readmission rate differences.
Response:
Thank you for these valuable insights. We have discussed the differences in healthcare systems between Japan and the United states and their relevance to the study results in the Discussion section.
- Discussion section would be easier to read and process if there were 4 separate paragraphs for each of the main variables you are discussing: BMI, Cognitive function, polypharmacy, & dependency
Response:
Thank you for the valuable suggestion. We have structured the Discussion into separate paragraphs for each of the main variables; cognitive function at discharge and dependency have been discussed in one paragraph for simplicity.
- Line 274: The article states "our result shows that rural older people with higher BMI have better nutritional conditions" is not correct. Albumin is a very poor predictor of malnutrition. This study was not designed to prove that hypothesis. Please rephrase this.
Response:
Thank you for pointing this out. We have eliminated the expression regarding albumin in the revised manuscript.
- Line 288: The article states "most patients who are readmitted had dehydration". I do not believe this is accurate as the table for only indicates 5.1% of all readmissions were from dehydration. Please consider rephrasing this.
Response:
We apologize for the misleading information. We have rephrased the sentence by stating that 13.9% of the readmitted patients had dehydration or a heat stroke in the revised manuscript.
Title, author list, abstract, ethics, references:
Title: Title of the article should be changed to reflect the singular hospital that this study involved. It says "risk of hospital readmission among older patients discharge from rehabilitation units in rural community hospitals". There was only one hospital involved in the study.
Response: Thank you for pointing this out. We have revised the Title to reflect that only one hospital was involved in this study.
Abstract:
- Line 25-27: The article needs to be clear that this type of study cannot differentiate between causation and correlation. The abstract states that the variables that they discovered "helped lower the rate of readmission" and that addressing these variables "will help improve outcomes". This is similar in the conclusion line 315 in which the article says the 4 variables “lower the risk of readmission”. Although I share the optimism that addressing these variables is the right approach the language needs to be changed to reflect this is a correlation rather than a causation.
Response:
Thank you for pointing this out. In accordance with your suggestions, we have toned down the conclusions in the revised manuscript.
- Keywords: Consider adding more keywords to help with search engine optimization: geriatrics, BMI, FMI, ADL?
Response:
Thank you for the suggestions; we have modified the keywords accordingly.
Ethics: Line 329-330 Informed consent statement states that informed consent was obtained from all subjects involved in the study. Is this a correct statement? That would mean there would have to be written informed consent from all 731 participants. Maybe that is part of routine admission to this hospital (obtaining a blanket informed consent statement for research) but usually one needs IC done separately from clinical hospital treatment release.
Response:
We apologize for the lack of clarity. The information related to this study was posted on the hospital website without the disclosure of any details concerning the patients. To address any questions regarding this study, contact information of the hospital representative was also listed on the website. The purpose of this study was explained to all patients, and informed consent was obtained. We have specified these details in the revised manuscript.
Round 2
Reviewer 1 Report
The authors conducted considerable revision and addressed the concerns of the reviewers, resulting in an excellent manuscript that is ready for acceptance.
All concerns from this reviewer in round 1 have been addressed.
Reviewer 2 Report
The authors have made the changes I suggested and I have no further changes or edits.